# Occupational Psychosocial Risks and Quality of Professional Life in Service Sector Workers with Sensory Processing Sensitivity

**DOI:** 10.3390/bs13060496

**Published:** 2023-06-13

**Authors:** Antonio Chacón, María Luisa Avargues-Navarro, Manuela Pérez-Chacón, Mercedes Borda-Mas

**Affiliations:** 1Spanish Association of Highly Sensitive Professionals and Psychologists, PAS España, 28080 Madrid, Spain; manuela.perez@pasespana.org; 2Department of Personality, Assessment, and Psychological Treatment, University of Seville, 41018 Seville, Spain; avargues@us.es

**Keywords:** highly sensitive persons, stress, burnout, compassion fatigue, compassion satisfaction, prevention

## Abstract

The aim of this study was to analyze the role of sensory processing sensitivity in the perception of stress under certain working conditions and its relationship with indicators of quality of professional life, in service sector workers. The participants (*n* = 3180) completed the Spanish versions of HSPS-S, CoPSoQ and ProQoL. The results show that exposure to certain working conditions represents a risk to the quality of professional life in workers of different fields, such as education, healthcare, hospitality and administration/management. The presence of high sensitivity is associated with poorer quality of professional life, specifically burnout and compassion fatigue. This study demonstrates the need to develop prevention programs aimed at managing stress by improving the working conditions, in order to adequately address sensory processing sensitivity and, consequently, promote the quality of professional life of service sector workers who present high sensitivity.

## 1. Introduction

In recent years, certain working conditions, such as job demands, sense of usefulness, decision management, decision making, activity organization and concerns and aspects derived from interpersonal relationships, are the main causes of potentially harmful processes that generate risks to workers’ health. In this sense, work-related stress is one of the major occupational health risks that affect workers of all sectors [1], especially in those professions in which physical and psychological tensions are greater, due to the interaction with customers [2].

Recent studies based on meta-analyses show that certain conditions of the working environment can affect professional development [3], job satisfaction [4] and the prevalence of alterations in physical and mental health [4,5], with an increase in symptoms of depression [6] and stress-related mental disorders [7].

From an interactionist model, stress is experienced from the exposure of the worker to the working conditions, being greater when an imbalance is perceived between the demands of the job and the available resources to meet those demands, affecting the worker’s well-being [8,9]. In this sense, the stress experienced in the workplace is influenced by a set of personal variables. Under similar working conditions, emotional responses differ between people. This difference in sensitivity is identified in the population from the concept of highly sensitive persons, such as those whose sensory processing sensitivity (SPS) to environmental stimuli may interfere in their daily living [10], as they are different from people without high sensitivity in the capacity to process the stimuli received from both positive and negative environmental influence [11,12]. In this sense, people with SPS, i.e., around 20% of the world population [11], due to their phenotypical temperament characteristics, are easily overwhelmed by both external stimuli (e.g., psychosocial risk factors) and internal stimuli (e.g., thoughts, perception of thoughts, emotion generated by thoughts, etc.) [13].

Chronic exposure to work-related stress factors, such as long shifts and workplace tension, has negative effects on the health of people with SPS [14], affecting their quality of professional life. The indicators of quality of life used in the present study were those aspects that appeared most frequently in the professionals who provide care or services to other people. Following Morante-Benadero et al. [15], quality of professional life is that which is perceived in the workplace as a help professional, and it is influenced by positive results (compassion satisfaction) and negative consequences (compassion fatigue and burnout). In fact, previous studies have reported the presence of burnout and compassion fatigue in professionals at the workplace [16], showing compassion fatigue as an emerging psychosocial risk that has recently become more severe, with different researchers studying it in the fields of education and healthcare [17], especially in those professions exposed to trauma [18], being a potential vulnerability factor for the development of compassion fatigue, according to Figley’s model [19].

Moreover, these indicators of quality of life acquire greater relevance among service sector workers, considering, in addition, the existence of the high-sensitivity trait in people. Workers with SPS stand out in certain resources, such as empathy, to cope with stress and other adverse working conditions. However, empathy may alter the perception of stress in highly sensitive workers, in one way or another, depending on the type of relationship they establish in their professions. Different studies show that people with high levels of empathy express compassion satisfaction [20]; nevertheless, SPS appears in empathic people [21,22,23] with greater suffering, as they perceive other people’s suffering and care for their well-being, especially in those professions that involve a direct relationship between the professionals and the users to attend to a basic need, such as health (healthcare sector) and education (education sector). Pérez-Chacón et al. [17] demonstrate the involvement of empathy in sectors such as education and healthcare, reporting that some people with SPS are more susceptible to burnout, compassion fatigue and compassion satisfaction. This suggests that the empathy of highly sensitive people could be a factor that affects the development of compassion fatigue or burnout, or, on the contrary, it may influence compassion satisfaction, thus impacting the quality of life. On the other hand, the literature does not show any studies on other service sector professions in which suffering is associated with other needs, such as well-being (nutrition and rest) (hospitality sector) and the management of economic matters (administration/management sector), which is an element of interest for the authors of the present study.

Regarding the above mentioned, some people with SPS are more susceptible to suffering from stress, depression and anxiety [24,25] with long-term consequences in the quality of professional life due to the risks of exposure to certain working conditions [16,26], burnout and/or compassion fatigue. In order to advance in the research on this topic, it is necessary to analyze the risk or protection factors, with the aim of identifying, in different working environments related to the service sector, how SPS may be affecting the perception of stress in the face of different working conditions.

The present study is focused on analyzing the potential role of SPS in the perception of stress under certain working conditions and its relationship with indicators of quality of professional life in service sector workers. Specifically, four objectives were proposed: (1) to determine, in different activity sectors (education, healthcare, hospitality, administration/management), the stress experienced under different working conditions (psychological demands, control over work and possibilities of development, social support and quality leadership, compensations and double presence), as well as the levels of quality of life in the workplace (burnout, compassion fatigue and compassion satisfaction) and SPS, exploring the presence of differences between the different sectors; (2) to analyze, in each sector, the differences in the psychosocial risk factors and indicators of quality of professional life, as a function of the SPS level presented by the workers (low, medium or high); (3) to determine the relationship between stressful working conditions and the indicators of quality of professional life in highly sensitive workers; and (4) to explore, in the workers with high sensory processing sensitivity, how the exposure to stressful psychosocial factors affects the indicators of quality of professional life, and whether there are variations as a function of the sector that the workers belong to. 

Based on the objectives of this study, it is expected that exposure to certain working conditions may pose a risk to the quality of working life of workers in various service sector occupations. Specifically, it is expected that high psychological demands, lack of social support and quality leadership and difficulty in finding a work–family balance act as risk factors, while control at work and development possibilities at work play a protective role. In addition, the presence of high sensitivity is expected to be associated with a higher presence of the negative dimensions of quality of work life (burnout and compassion fatigue).

The aim of the present study was to determine how SPS influences workers in their working environment, both regarding the perception of certain working conditions as stressful and in terms of the presence of indicators of their quality of professional life, focusing on professionals who provide care or services, and exploring the existence of variations as a function of their specific sector. The results of this study are expected to guide the development of programs on stress management, prevention of burnout and compassion fatigue and promotion of compassion satisfaction. Knowing the working conditions and sectors in which workers with SPS are more or less vulnerable could help to develop efficient coping resources for them. 

## 2. Materials and Methods

### 2.1. Participants

The initial sample consisted of 10,525 Spanish adults (1741 men and 8784 women) with an average age of 33.61 years (SD = 11.40) (range 18–72 years), who were recruited by convenience sampling, through proximity in the community context. A total of 3180 participants met the following inclusion criteria: (a) being at least 18 years old, (b) adequately completing all the data and tests and (c) providing informed consent. 

The final sample of this study was constituted by workers of the service sector (education 37.42% *n* = 1190, healthcare 28.74% *n* = 914, hospitality 9.59% *n* = 305 and administration/management 24.24% *n* = 771), with an average age of 37.41 years (SD = 10.21) (range 18–72 years). Regarding their employment situation, 2498 (78.56%) were employed and 682 (21.44%) were self-employed, with an average of 7.55 years (SD = 8.05) in their job at the time of the study. With regard to sex, 2784 (87.55%) were women and 393 (12.45%) were men. Table 1 presents the main sociodemographic and job characteristics of the participants as a function of the sector they belong to.

### 2.2. Procedure

The sample was recruited between April and May 2020. To this end, we first contacted the population of highly sensitive people, associations interested in the topic and professionals of Spanish universities for the dissemination of the information about the study. Then, the online anonymous tests, which lasted between 15 and 20 min, were carried out through a web-based application. After reading a brief introduction with the objectives of the study, the candidates agreed to participate under the research conditions proposed. Subsequently, the tests were completed, always in the same order. 

Participation was voluntary, anonymous and non-remunerated. All participants signed their informed consent form, which stated that they could leave the study whenever they wished to. The necessary measures were taken to safeguard the information in compliance with Organic Law 3/2018 on the protection of data and guarantee of digital rights. 

The study was conducted following the ethics code of the World Medical Association Declaration of Helsinki [27], Psychologist Code of Ethics (Código Deontológico Del Psicólogo) [28] and the ethical principles recommended for research with human participants [29]. 

### 2.3. Data Analysis

The statistical analyses were conducted using the Statistical Package for Social Sciences for Windows (SPSS), v26.0 [30].

Descriptive analyses (frequencies, percentages, means and standard deviations) were conducted for the different variables for the four groups of professionals. The normality assumption was tested with the Kolmogorov–Smirnov test, and the assumption of homoscedasticity was verified with Levene’s test. To determine the differences between variables as a function of the working sector and the levels of SPS, single-factor ANOVAs (Fisher’s F or Kruskal–Wallis H) were conducted, and Student’s *t*-tests or Mann–Whitney U-tests were carried out for post hoc comparisons, depending on whether or not the requirement of variance homogeneity was met. Likewise, Cohen’s d was calculated using Lipsey and Wilson’s formula, considering the sample sizes in the four groups. The reference values were <0.30, 0.30–0.50 and >0.50 for small, medium and large sizes, respectively. Pearson’s r correlation analysis was performed for the relationship between working conditions and quality of professional life in the workers with medium and high levels of SPS for each sector. The values of the effect size were small (<0.30), medium (0.30–0.49) and large (>0.49). Lastly, to determine the influence of the working conditions on the indicators of quality of professional life (burnout, compassion fatigue and compassion satisfaction) in the workers with medium–high SPS, a multiple-regression analysis was conducted using the “enter” method for each sector.

### 2.4. Measures

#### 2.4.1. Ad Hoc Questionnaire of Sociodemographic and Job Information

This consists of 12 questions about sex, age, marital status, education level, number of children, place and autonomous community of residence, profession, professional sector, job position and time in the current job. 

#### 2.4.2. High-Sensitivity Person Scale (HSPS): HSPS-S, Spanish Adaptation by Chacón et al. 

HSPS-S [31] is a self-reported scale to identify highly sensitive people. It consists of 27 direct items with seven response options in a Likert scale (1 = totally disagree/7 = totally agree) (range: 27–189). Higher scores indicate a greater degree of sensory processing sensitivity (SPS). In its original version [10], the internal consistency coefficients were α = 0.87 and α = 0.85. In the present study, the Cronbach’s α and McDonald’s omega coefficients for the different subscales were: α = 0.87 and ω = 0.87 for sensitivity overstimulation (SOS: 5, 11, 14, 16, 19, 21, 23, 26 and 27) (feeling of being overwhelmed by the external and internal demands), α = 0.80 and ω = 0.80 for aesthetic sensitivity (AES: 2, 3, 8, 10, 15 and 22) (awareness of the surrounding aesthetics), α = 0.83 and ω = 0.84 for low sensory threshold (LST: 1, 7, 9, 18 and 25) (sensory discomfort due to overstimulation), α = 0.57 and ω = 0.59 for fine psychophysiological discrimination (FPD: 4, 6, 13 and 20) (discrimination of subtleties or physical/physiological sensitivity in response to internal stimuli) and α = 0.70 and ω = 0.71 for harm avoidance (HA: 12, 17 and 24) (controlled avoidance of damage). For the total HSPS-S scale, α = 0.92 and ω = 0.92 were obtained. In this study, the global score of HSPS-S was used. 

#### 2.4.3. ISTAS 21 (CoPSoQ) Copenhagen Psychosocial Questionnaire, Created in Denmark by Kristensen et al.: Spanish Version: ISTAS 21, Moncada et al.

The Copenhagen Psychosocial Questionnaire [32] evaluates psychosocial risks in the job context, in the Spanish version [33] specifically the worker’s risk perception level (low, medium and/or high), based on the demand–control–social support (DCSS) by Karasek [9] and effort–reward imbalance (ERI) of Siegrist´s models [34]. Its short version was used in this study, and it consists of 20 items, with five Likert response options (0 = never/4 = always), grouped in five dimensions: (a) psychological demands (PD: 1, 2, 3, 4 and 5; range 0–20) (it measures both the demand and the effort of the worker), (b) control over work and possibilities of development (CW-PD: 6, 7, 8, 9 and 10; range 0–20) (control equivalent; it evaluates the autonomy of the worker), (c) social support and quality leadership (SS-QL: 11, 12, 13, 14 and 15; range 0–20) (social support equivalent; it contains leadership elements), (d) compensations (C: 16, 17 and 18; range 0–12) (reward, recognition equivalent; it measures the effort–reward imbalance, as well as the worker status control) and (e) double presence (DP: 19 and 20; range 0–8) (interference or work–family conflict equivalent; it evaluates the concern for simultaneously fulfilling the household and job tasks). Higher scores indicate a higher psychosocial risk situation. In items 1, 6, 7, 8, 9, 10, 11, 13, 14, 15 and 18, the score is inverted. The classification of low, medium and high score is: PD (low: 0–8; medium: 9–11; high: 12–20), CW-PD (low: 0–5; medium: 6–8; high: 9–20), SS-QL (low: 0–3; medium: 4–6; high: 7–20), C (low: 0–2; medium: 3–5; high: 6–12) and DP (low: 0–1; medium: 2–3; high: 4–8). In this study, the Cronbach’s α and McDonald´s omega coefficients for the subscales were: PD (α = 0.58; ω = 0.71), CW-PD (α = 0.64; ω = 0.69), SS-QL (α = 0.70; ω = 0.75), C (α = 0.55; ω = 0.62) and DP (α = 0.64; ω = 0.65). All five dimensions were used.

#### 2.4.4. Spanish Adaptation of the Professional Quality of Life Scale (ProQoL-vIV) by Morante-Benadero et al.

The Spanish adaptation of ProQOL-vIV [15] was used to evaluate compassion fatigue, compassion satisfaction and burnout. This scale consists of 30 items, which measure positive and negative aspects of empathy in professionals. The items have six response options (0 = never/5 = always). It includes three dimensions: compassion fatigue (CF: items 2, 5, 7, 9, 11, 13, 14, 23, 25 and 28), compassion satisfaction (CS: items 3, 6, 12, 16, 18, 20, 22, 24, 27 and 30) and burnout (BU: items 1, 4, 8, 10, 15, 17, 19, 21, 26 and 29). In items 1, 4, 15, 17 and 29, the score is inverted. The classification of low, medium and high score is: CF (low: 0–7; medium: 8–17; high: 18–50), CS (low: 0–32; medium: 33–41; high: 42–50) and BU (low: 0–17; medium: 18–27; high: 28–50). The reliability of the scale was 0.71 and the internal consistency coefficients were: CF (α = 0.80; ω = 0.81), CS (α = 0.85; ω = 0.85) and BU (α = 0.64; ω = 0.67). All three dimensions were used.

## 3. Results

### 3.1. Differences between Professional Sectors

The first objective of this study was to determine the mean scores in the different study variables for each sector, as well as the existence of significant differences between sectors. In sensory processing sensitivity (HSPS-S), the highest mean score was observed in education (154.99), followed by hospitality (147.43), healthcare (145.49) and administration/management (144.07) (Table 2). In all four professional sectors, the women obtained higher mean scores (hospitality = 158.41; education = 157.18; administration/management = 156.81; healthcare = 153.86) than the men (hospitality = 147.43; education = 154.99; administration/management = 144.07; healthcare = 145.49), with significant differences (*p* = 0.001), except in education (*p* = 0.256). Regarding sectors, significant differences were detected between education and healthcare (*p* = 0.000), with higher scores in education (Table 3), obtaining a small effect size.

With respect to the indicators of quality of professional life (ProQoL), it should be noted that, in CF, the hospitality workers showed the highest mean score (24.35), followed by the education workers (22.37), administration/management workers (22.22) and, lastly, the healthcare workers (20.98). In this case, significant differences were found in the mean scores between education and healthcare (E > HE) (*p* = 0.001) and between education and hospitality (E > HO) (*p* = 0.001), as well as for hospitality with healthcare (HO > HE) (*p* = 0.000) and administration/management (A/M > HO) (*p* = 0.009) and between hospitality and administration/management (HO > A/M) (*p* = 0.001). All effect sizes were small.

In BU, the hospitality workers obtained the highest mean score (25.70), followed, in this case, by the administration/management workers (25.33), the education workers (24.07), and, lastly, the healthcare workers (23.33). The significant differences were established for the mean scores of hospitality from those of education (HO > E) (*p* = 0.001) and healthcare (HO > HE) (*p* = 0.000), as well as for the mean scores of administration/management and education (A/M > E) (*p* = 0.000) and healthcare (A/M > HE) (*p* = 0.000). All effect sizes were small.

Finally, in CS, the education and healthcare workers presented the highest mean scores (37.91 and 37.71, respectively), whereas the hospitality workers obtained the lowest mean scores (31). Moreover, it is worth pointing out that the mean scores of the latter were significantly different from those obtained in the rest of the sectors: education (HO < E) (*p* = 0.000), healthcare (HO < HE) (*p* = 0.000) and administration/management (HO < A/M) (*p* = 0.037). Likewise, significant differences were found for the mean scores of administration/management from those of education (A/M < E) (*p* = 0.000) and healthcare (A/M < HE) (*p* = 0.000), with the score of administration/management being lower in both cases. The effect sizes were medium, except for the difference between HO and A/M, where the effect size was small.

Regarding the different psychosocial risks or stressful working conditions, the healthcare workers presented the highest scores in PD (12.10), followed by the hospitality workers (11.79), the education workers (11.47) and the administration/management workers (10.86). Significant differences were found for the mean scores of healthcare from those of education (HE > E) (*p* = 0.000) and administration/management (HE > A/M) (*p* = 0.000). Likewise, significant differences were also found for administration/management from education (E > A/M) (*p* = 0.000) and hospitality (HO > A/M) (*p* = 0.000). All effect sizes were small.

In the case of CW-PD, the hospitality workers presented the highest mean scores (9.98), followed by administration/management (8.83), healthcare (6.93) and education (6.91), obtaining significant differences for hospitality from education (HO > E) (*p* = 0.000), healthcare (HO > HE) (*p* = 0.000) and administration/management (HO > A/M) (*p* = 0.000), with a medium effect size in the first case and a small effect size in the other two cases. Similarly, significant differences were obtained for administration/management from education (A/M > E) (*p* = 0.000) and healthcare (A/M > HE) (*p* = 0.000), with small effects sizes.

In SS-QL, the administration/management workers (8.13) obtained the highest scores, followed by the hospitality workers (8.01), the healthcare workers (7.91) and the education workers (7.84), although no significant differences were detected in the mean scores between sectors. 

In regard to the mean scores of C, it was again the hospitality workers who presented the highest values (5.26), followed by the healthcare workers (5.18), the administration/management workers (4.96) and the education workers (4.64), detecting significant differences for the scores of hospitality from those of education (HO > E) (*p* = 0.003) and healthcare (HO > HE) (*p* = 0.001), as well as for healthcare from administration/management (HE > A/M) (*p* = 0.011). All effect sizes were small.

Lastly, in DP, it was the administration/management workers who presented the highest scores (3.29), followed by the education workers (3.01), the hospitality workers (2.94) and the healthcare workers (2.92), showing significant differences from the other three sectors (A/M > E, *p* = 0.010; A/M > HO, *p* = 0.004; A/M > HE, *p* = 0.000). All effect sizes were small.

### 3.2. Differences between Professional Sectors as a Function of SPS

The second objective of this study was to explore, in each sector, the existence of differences in the psychosocial risk factors and indicators of quality of life, as a function of the SPS level of the workers.

With respect to the indicators of quality of professional life as a function of the SPS levels, as can be observed in Table 4, regardless of the sector, the highest mean scores in the three subscales (CF, BU and CS) were obtained by high SPS level. In all sectors, in the high SPS level, a high risk level was observed in CF, and a medium risk level was detected in BU and CS, except in administration/management workers, where the risk level was low–medium. However, the risk level in quality of professional life was similar, in general, to that of the low and medium levels of the four sectors. 

Considering the significant differences between the mean scores presented for each level in each sector, Table 5 shows that, in the case of CF, there were significant differences between all levels in all four sectors (*p* = 0.000), with a medium and large effect size in the comparisons between the low and high levels. With regard to BU, while education showed differences between all groups (*p* = 0.000), in the case of healthcare, the differences were found between the low-level and high-level groups (*p* = 0.004). In hospitality, the differences were detected between the low-level and medium-level groups (*p* = 0.000), with a medium effect size. Lastly, in the case of administration/management, the differences were found between all groups, although with different significance (low–medium and medium–high: *p* = 0.014; low–high: *p* = 0.000). Lastly, in relation to CS, significant differences were only found in hospitality between the low-SPS and high-SPS groups, and between the medium-SPS and high-SPS groups (*p* = 0.000 in both cases), with small effect sizes. 

In regard to the working conditions, as can be observed in Table 4, in general, the scores in each of the stressful working conditions were higher in the high-SPS group, regardless of the sector. In all sectors and for the different psychosocial factors, the high-SPS group presented a high risk level (in SS-QL; in hospitality in CW-PD; and in PD, except in administration/management) and a medium–high level (in C; in DP; and in administration/management in PD and CW-PD). Moreover, to a lesser extent, the level was medium in education and healthcare in CW-PD.

Taking into account the comparisons between groups in each sector, Table 5 shows that, in the case of education, there were significant differences between the mean scores obtained in the three levels in PD (*p* = 0.000). No significant differences were detected in CW-PD (*p* = 0.534). In SS-QL, differences were found between the low-level and high-level groups (*p* = 0.036) and between the medium-level and high-level groups (*p* = 0.004). The same was observed in C, obtaining significant differences between the high-level and low-level groups (*p* = 0.000), as well as between the medium-level and high-level groups (*p* = 0.003). Lastly, as can be observed, significant differences were detected between all groups in the case of DP (*p* = 0.000). It is worth highlighting the medium effect size in the comparisons between low and high levels in PD and DP.

Regarding the healthcare workers, significant differences were found in PD between the low-level and high-level groups (*p* = 0.000). As in the case of the education workers, no significant differences were found in CW-PD (*p* = 0.719). With regard to SS-QL, significant differences were only found between the mean scores of the low-SPS and high-SPS groups (*p* = 0.001). In the case of C, differences were detected between all groups: low–medium (*p* = 0.016), low–high (*p* = 0.000) and medium–high (*p* = 0.020). Lastly, in DP, significant differences were observed between the low-SPS and high-SPS groups (*p* = 0.000), as well as between the medium-SPS and high-SPS groups (*p* = 0.002). All effect sizes were small.

In the hospitality workers, significant differences were only found for PD, C and DP. In the case of PD, the differences were obtained between the low-SPS and medium-SPS groups (*p* = 0.006) and between the low-SPS and high-SPS groups (*p* = 0.000). In C and DP, the differences were detected between the low-SPS and high-SPS groups (*p* = 0.012 and *p* = 0.008, respectively). All effect sizes were small.

Lastly, in the administration/management workers, in the case of PD, significant differences were found between all levels (low–medium: *p* = 0.015; low–high: *p* = 0.000 and medium–high: *p* = 0.033). In CW-PD, a significant difference was found only between the low-level and high-level groups (*p* = 0.014), with no significant differences in SS-QL (*p* = 0.332). Finally, it is worth pointing out that in both C and DP, significant differences were detected between the low-SPS and medium-SPS groups (C: *p* = 0.011; DP: *p* = 0.002), as well as between the low-SPS and high-SPS groups (C: *p* = 0.040; DP: *p* = 0.000). All effect sizes were small.

### 3.3. Relationship between SPS, Working Conditions and Quality of Professional Life

The third objective of this study was to explore the relationship between the different working conditions and the indicators of quality of professional life in workers with a medium–high SPS level. To this end, correlation analyses (r) were performed in each sector. As can be observed in Table 6, in all sectors, the different working conditions were directly correlated with CF and BU and inversely correlated with CS. All correlations were significant, except for the one established between DP and CS in education (*p* = 0.163), hospitality (*p* = 0.064) and administration/management (*p* = 0.446). In the case of hospitality, the correlation between CW-PD and CF was not significant (*p* = 0.158).

It is important to highlight the direct correlations of PD with BU in education (r = 0.50), hospitality (r = 0.53) and administration/management (r = 0.55), the correlation established between SS-QL and BU in healthcare (r = 0.50) and the inverse correlation between CW-PD and CS in all sectors (education r = −0.53; healthcare r = −0.64; hospitality = −0.56; administration/management r = −0.55), with large effect sizes.

### 3.4. Predictive Value of the Working Conditions on the Indicators of Quality of Professional Life in Highly Sensitive Persons

Finally, to study the influence of the psychosocial risk factors on the indicators of quality of professional life in the workers with medium–high sensitivity in each sector, multiple-regression analyses were performed, using the successive steps method. For these analyses, the different working conditions were considered independent or predictor variables, and each of the indicators of quality of professional life (CF, BU CS) was considered a dependent or criterion variable. 

From the proposed models, in the education workers, the percentages of variance explained in the indicators of quality of professional life were 34.3% in CF (F = 111.68), 44.1% in BU (F = 134.82) and 30.2% in CS (F = 123.41). In the healthcare workers, the percentages of variance explained were 35.1% in CF (F = 104.02), 45.3% in BU (F = 95.33) and 42.3% in CS (F = 212.31). In the hospitality workers, the percentages were 33.1% in CF (F = 35.75), 39.8% in BU (F = 35.69) and 34.2% in CS (F = 56.63). Lastly, in the administration/management workers, the percentages were 31.7% in CF (F = 80.88), 40.4% in BU (F = 88.77) and 33.5% in CS (F = 87.91). In all regression analyses, the *p*-value was *p* = 0.000 (Table 7).

In the different models for CF, in education (β = 6.191; t = 6.545; *p* = 0.000), healthcare (β = 2.983; t = 6.545; *p* = 0.017), hospitality (β = 8.987; t = 4.731; *p* = 0.000) and administration/management (β = 8.334; t = 7.765; *p* = 0.000), the results indicate that the working conditions related to PD (*p* = 0.000), DP (*p* = 0.000) and C (*p* = 0.000) would act as risk factors in the four professional sectors, as well as SS-QL (*p* = 0.000) in education (Table 7).

Regarding the different models for BU, in education (β = 6.880; t = 9.051; *p* = 0.000), healthcare (β = 4.570; t = 4.450; *p* = 0.000), hospitality (β = 8.051; t = 4.701; *p* = 0.000) and administration/management (β = 10.287; t = 11.546; *p* = 0.000), the results indicate that the working conditions related to PD (*p* = 0.000), CW-PD (*p* = 0.000) and C (*p* = 0.000, education and healthcare; *p* = 0.001, hospitality; *p* = 0.003, administration/management) would act as risk factors. Likewise, DP in three sectors (*p* = 0.000, education; *p* = 0.005, healthcare; *p* = 0.003, hospitality) and SS-QL in three sectors (*p* = 0.000, education and healthcare; *p* = 0.002, administration/management) would also act as risk factors.

Finally, in the models for CS, in education (β = 4.801; t = 77.987; *p* = 0.000), healthcare (β = 51.194; t = 43.107; *p* = 0.000), hospitality (β = 46.049; t = 24.538; *p* = 0.000) and administration/management (β = 45.134; t = 40.147; *p* = 0.000), the results indicate that the working conditions related to CW-PD (*p* = 0.000) in the four sectors, PD in healthcare (*p* = 0.000), hospitality (*p* = 0.003) and administration/management (*p* = 0.016), C in education and administration/management (*p* = 0.000) and SS-QL in education (*p* = 0.022) would act as protective factors on compassion satisfaction.

## 4. Discussion

Personality traits and job characteristics are important factors to determine the individual health state at the workplace [35]. The general aim of this study was to analyze the role of SPS in the perception of stress under certain working conditions and its relationship with indicators of quality of professional life. The novelty of the present study lies in the fact that it was simultaneously conducted with workers of different professions of the service sector, specifically, education, healthcare, hospitality and administration/management. The results obtained have allowed us to confirm our initial assumptions.

With regard to the perceived stress level, it was observed that the workers of the four sectors were exposed to certain work-related psychosocial risks, concretely between 28.7% (CW-PD) and 37.4% (SS-QL) in education, between 23.6% (CW-PD) and 31.8% (PD) in healthcare and between 21.7% (PD) and 32% (CW-PD) in administration/management. On the other hand, in hospitality, the highest percentage was 15.7% (in C and CW-PD), which is partly due to the fact that the working conditions in this sector lead to consideration of it as a transition to other sectors [36]. However, it is worth highlighting that, for the education workers, the stress factor of greatest risk was the one related to social support and quality leadership, whereas for the healthcare workers, the greatest risk was found in psychological demands, and for the workers in hospitality and administration/management, the highest risk was obtained in the factors related to control over work and possibilities of development; these findings are in line with the proposition of Karasek’s model [9] about the influence of these demands on the appearance of work-related stress. Therefore, identifying the working conditions implies a central interest for the management of stress and health [35] in the workplace.

The results of this study show that the stress perceived in the work environment, specifically in service sector workers, affects the quality of professional life. In this sense, the presence of the different stressful factors is associated with the presence of burnout and compassion fatigue, with a more remarkable effect in education and healthcare. These results are consistent with those of previous studies, which report that the different working conditions and the nature of the job itself influence the health of the workers [37]. Furthermore, different studies assert that people with high sensory processing sensitivity are more susceptible to burnout and compassion fatigue [17].

In line with previous research on stress and health [16], and in relation to stress and sensory processing sensitivity [38], in this study, the stress caused by the working conditions favors the appearance of burnout and compassion fatigue, with both becoming worse with higher sensitivity, in the participants of all four sectors analyzed. It is worth pointing out that, among the different working conditions, psychological demands, social support and quality leadership and family–work balance were the factors that showed risk for the presence of burnout and compassion fatigue in people with medium–high sensitivity. In line with these findings, previous studies indicate that sensory processing sensitivity is associated with burnout and compassion fatigue [39] and that the burnout rate varies among the different professions and empathy is associated with factors that include working conditions [40].

Regarding the relationship between compassion satisfaction and quality of professional life, in general, the obtained findings show that this was similarly present in the different sectors, as well as in the participants as a function of the levels of sensory processing sensitivity. Nevertheless, it is worth highlighting that greater compassion satisfaction was associated with better working conditions, that is, with lower perception of work-related stress. Of the different working conditions, having greater control over work and possibilities for development was the most relevant protection factor for compassion satisfaction in the participants with medium–high sensitivity for the different sectors analyzed. As was previously mentioned, people with high levels of empathy, in addition to compassion fatigue, also presented compassion satisfaction [19], especially in education and healthcare. The findings of this study show, as a novelty, that, in a similar manner for the different professions of the service sector, being able to understand, attend to and respond to the needs of the clients/users with certain autonomy and the possibility of developing skills, in the exercise of their occupation, are protection and enrichment factors for their quality of professional life.

In summary, this study demonstrates the influence of the perception of stress under different working conditions on the quality of professional life, through the indicators of health (burnout and compassion fatigue) and compassion satisfaction in people with sensory processing sensitivity who work in the service sector.

## 5. Conclusions

Based on the objectives set for this study, the results show that exposure to certain working conditions is a risk to the health of workers of different professions, specifically in the service sector, such as education, healthcare, hospitality and administration/management. Moreover, the presence of high sensitivity is associated with a worse quality of professional life, greater presence of burnout and compassion fatigue, especially in education and healthcare. Moreover, psychological demands, social support and quality leadership and the difficulty of finding family–work balance are working conditions with greater predictive power, acting as risk factors for the appearance of burnout and compassion fatigue. In turn, control over work and possibilities for development act as protection factors on compassion satisfaction.

These findings show the need to develop prevention programs aimed at planning and improving working conditions that allow enhancing the management of stress for the adequate control of sensory processing sensitivity and, consequently, promoting the quality of professional life, especially in workers of different professions related to the service sector who present high sensitivity, as is stated by [35].

These results are very useful in proposing preventive actions for people with high sensitivities. However, this study has some limitations that must be pointed out. Firstly, we used a short version of the questionnaire on working conditions, with a small number of items in some of the dimensions, which implies that, despite the fact that it gathers those items with greater factor loadings, the adaptation and interpretation of the results may be affected depending on the social group that responds to the questionnaire. Secondly, the study excluded the participants who did not have access to the Internet, favoring, on the one hand, the spread of the information and participation of the population, and, on the other hand, discarding people with these characteristics whose information would have been beneficial for the generalization of the findings of the present study. Future works should replicate this study with a different instrument to compare these findings, expanding the evaluation by including the face-to-face modality, in order to delve into this topic and guarantee equal opportunities for the target study population, especially for the sake of the adequate implementation of prevention programs. Thirdly, we did not use a qualitative methodology along with, or replacing, the quantitative methodology, which would have allowed contributing greater knowledge and dissemination to the scientific community from a different perspective. Lastly, it is important to be aware of the predictive limitations of cross-sectional studies, such as the fact that there is usually no evidence of a temporal relationship between exposure and outcome. Without longitudinal data, it is possible to make a mistake when attempting to establish a true cause–effect relationship.

## Figures and Tables

**Table 1 behavsci-13-00496-t001:** Characteristics of the study participants (*n* = 3180).

		Education (*n* = 1190)		Healthcare (*n* = 914)		Hospitality (*n* = 305)		Administration /Management (*n* = 771)	
		F	%	F	%	F	%	F	%
Sex and age									
	Women (*n* = 2784)	1048	33	817	25.7	249	7.8	670	21.1
	Mean age (Range)	37.16 18–68		37.56 20–70		30.62 18–60		38.90 19–65	
	SD	9.78		9.80		8.44		9.97	
	Men (*n* = 393)	142	4.5	97	3.1	56	1.8	101	3.2
	Mean age (Range)	38.84 18–70		42.85 19–70		31.39 18–70		41.86 18–72	
	SD	9.79		11.94		10.04		12.45	
Age group									
	≤30	349	11	241	7.6	174	5.5	194	6.1
	31–40	412	13	327	10.3	91	2.9	224	7
	41–50	306	9.6	219	6.9	27	0.8	224	7
	51–60	110	3.5	104	3.3	11	0.3	120	3.8
	≥61	13	0.4	23	0.7	2	0.1	9	0.3
Marital status									
	Single	442	13.9	309	9.7	166	5.2	262	8.2
	With partner	262	8.2	201	6.3	68	2.1	172	5.4
	Married	325	10.2	290	9.1	31	1	225	7.1
	Divorced	118	3.7	91	2.9	18	0.6	94	3
	Widowed	5	0.2	4	0.1	2	0.1	3	0.1
	Not specified	38	1.2	19	0.6	20	0.6	15	0.5
Education level									
	College	1041	32.7	736	23.1	123	3.9	509	16
	High school	140	4.4	165	5.2	136	4.3	224	7
	Secondary	6	0.2	11	0.3	38	1.2	34	1.1
	Primary	3	0.1	2	0.1	7	0.2	4	0.1
	Without studies	0	0	0	0	1	0	0	0
Type of contract									
	Employed	946	29.7	624	19.6	257	8.1	671	3.1
	Self-employed	244	7.7	290	9.1	48	1.5	100	31.65

**Table 2 behavsci-13-00496-t002:** Means and standard deviations by sectors in the study variables.

		Education (*n* = 1190)		Healthcare (*n* = 914)		Hospitality (*n* = 305)		Administration/ Management (*n* = 771)	
		M	SD	M	SD	M	SD	M	SD
Sensory Processing Sensitivity (SPS)									
		154.99	18.01	145.49	23.96	147.43	23.84	144.07	26.10
Professional Quality of Life (PQoL)									
	CF	22.37 ^(4)^	7.97	20.98 ^(4)^	8.06	24.35 ^(4)^	8.50	22.22 ^(4)^	8.11
	BU	24.07 ^(3)^	6.54	23.33 ^(3)^	6.80	25.70 ^(3)^	6.73	25.33 ^(3)^	6.68
	CS	37.91 ^(3)^	7.45	37.71 ^(3)^	7.73	31.00 ^(1)^	7.55	32.39 ^(2)^	7.96
Occupational Psychosocial Risks (OPR)									
	PD	11.47 ^(2)^	2.89	12.10 ^(3)^	2.79	11.79 ^(2)^	3.04	10.86 ^(1)^	3.41
	CW-PD	6.91 ^(1)^	3.27	6.93 ^(1)^	3.56	9.98 ^(3)^	4.11	8.83 ^(2)^	4.02
	SS-QL	7.84 ^(3)^	3.40	7.91 ^(3)^	3.73	8.01 ^(3)^	3.52	8.13 ^(3)^	3.83
	C	4.64 ^(1)^	2.73	5.18 ^(1)^	3.07	5.26 ^(1)^	2.87	4.96 ^(1)^	2.83
	DP	3.01 ^(1)^	1.98	2.92 ^(1)^	1.93	2.94 ^(1)^	1.90	3.29 ^(2)^	1.93

SPS: Sensory Processing Sensitivity; PQoL: Professional Quality of Life; CF: Compassion Fatigue; BU: Burnout; CS: Compassion Satisfaction; OPR: Occupational Psychosocial Risks; PD: Psychological Demands; CW-PD: Control over Work and Possibilities of Development; SS-QL: Social Support and Quality Leadership; C: Compensations; DP: Double Presence; PQoL: Level of exposure to indicators of professional quality of life (ProQoL): ^(1)^ low, ^(2)^ low–medium, ^(3)^ medium, ^(4)^ high. The high, medium and low score classification is: CF (low: 0–7; medium: 8–17; high: 18–50), BU (low: 0–17; medium: 18–27; high: 28–50) and CS (low: 0–32; medium: 33–41; high: 42–50). OPR: Level of exposure to occupational psychosocial risks (ISTAS21): ^(1)^ medium, ^(2)^ medium–high, ^(3)^ high. The high, medium and low score classification is: PD (low: 0–8; medium: 9–11; high: 12–20), CW-PD (low: 0–5; medium: 6–8; high: 9–20), SS-QL (low: 0–3; medium: 4–6; high: 7–20), C (low: 0–2; medium: 3–5; high: 6–12) and DP (low: 0–1; medium: 2–3; high: 4–8).

**Table 3 behavsci-13-00496-t003:** Comparison of means between Education, Healthcare, Hospitality and Administration/Management.

	Comparison of Intergroup Means			Paired Comparison of Mean (t/U)											
				Education						Healthcare				Hospitality	
				Healthcare		Hospitality		Administration/Management		Hospitality		Administration/Management		Administration/Management	
		F/H	*p*-Value	*p*-Value	Cohen’s d	*p*-Value	Cohen’s d	*p*-Value	Cohen’s d	*p*-Value	Cohen’s d	*p*-Value	Cohen’s d	*p*-Value	Cohen’s d
SPS		177.21	0.001 **	0.000 *** E > HE	0.45 s	0.297	0.36 s	0.108	0.49 s	0.075	−0.08 s	0.230	0.06 s	0.914	0.13 s
PQoL															
	CF	14.219	0.000 ***	0.001 ** E > HE	0.17 s	0.001 ** HO > E	−0.24 s	0.979	0.02 s	0.000 *** HO > HE	−0.41 s	0.009 ** A/M > HE	−0.16 s	0.001 ** HO > A/M	0.25 s
	BU	17.468	0.000 ***	0.057	0.11 s	0.001 ** HO > E	−0.24 p	0.000 *** A/M > E	−0.19 s	0.000 *** HO > HE	−0.35 s	0.000 *** A/M > HE	−0.30 s	0.842	0.05 s
	CS	139.895	0.000 ***	0.934	0.03 s	0.000 *** HO < E	0.92 h	0.000 *** A/M < E	0.72 m	0.000 *** HO < HE	0.87 m	0.000 *** A/M < HE	−0.68 m	0.037 * HO < A/M	−0.18 s
OPR															
	PD	24.496	0.000 ***	0.000 ** HE > E	−0.22 s	0.354	−0.11 s	0.000 *** E > A/M	0.19 s	0.387	0.11 s	0.000 *** HE > A/M	0.40 s	0.000 *** HO > A/M	0.29 s
	CW-PD	241.91	0.000 ***	0.885	−0.01 s	0.000 *** HO > E	−0.83 m	0.000 *** A/M > E	−0.52 s	0.000 *** HO > HE	−0.79 m	0.000 *** A/M > HE	0.50 s	0.000 *** HO > A/M	0.28 s
	SS-QL	1.075	0.000 ***	0.975	−0.02 s	0.874	−0.05 s	0.321	−0.08 s	0.971	−0.03 s	0.621	−0.06 s	0.962	−0.03 s
	C	7.303	0.000 ***	0.850	−0.18 s	0.003 ** HO > E	−0.22 s	0.059	−0.11 s	0.001 ** HO > HE	−0.03 s	0.011 * HE > A/M	0.07 s	0.380	0.10 s
	DP	192.96	0.000 ***	0.303	−0.05 s	0.515	0.04 s	0.010 * A/M > E	−0.14 s	0.958	−0.01 s	0.000 *** A/M > HE	−0.19 s	0.004 ** A/M > HO	−0.18 s

*** *p* ≤ 0.001, ** *p* ≤ 0.01. * *p* ≤ 0.05. p = significance level, F = Fisher; t = Student; H = Kruskal–Wallis; U = Mann–Whitney. Cohen’s d: effect size: s = small magnitude ratio: <0.30; m = mean: 0.30–0.50; h = high: >0.50. Healthcare = HE; Education = E; Hospitality = HO; Administration/Management = A/M. SPS: Sensory Processing Sensitivity. PQoL: Professional Quality of Life (ProQoL): CF: Compassion Fatigue; BU: Burnout; CS: Compassion Satisfaction. OPR: Occupational Psychosocial Risks (ISTAS21): PD: Psychological Demands; CW-PD: Control over Work and Possibilities of Development; SS-QL: Social Support and Quality Leadership; C: Compensations; DP: Double Presence.

**Table 4 behavsci-13-00496-t004:** Means and standard deviations according to the level of sensory processing sensitivity (SPS) in each profession.

		Education (*n* = 1190)			Healthcare (*n* = 914)			Hospitality (*n* = 305)			Administration/Management (*n* = 771)		
		Low ^(a)^ (*n* = 328)	Medium ^(b)^ (*n* = 393)	High ^(c)^ (*n* = 469)	Low (*n* = 332)	Medium (*n* = 307)	High (*n* = 275)	Low (*n* = 84)	Medium (*n* = 121)	High (*n* = 100)	Low (*n* = 244)	Medium (*n* = 256)	High (*n* = 271)
		M SD	M SD	M SD	M SD	M SD	M SD	M SD	M SD	M SD	M SD	M SD	M SD
PQoL													
	CF	17.58 ^(4)^ (6.60)	21.42 ^(5)^ (6.70)	26.52 ^(5)^ (7.68)	17.40 ^(3)^ (6.73)	20.85 ^(5)^ (7.34)	25.46 ^(5)^ (8.13)	19.04 ^(5)^ (7.81)	24.35 ^(5)^ (7.30)	28.82 ^(5)^ (7.88)	17.91 ^(4)^ (6.75)	22.59 ^(5)^ (7.63)	25.77 ^(5)^ (7.86)
	BU	22.08 ^(3)^ (6.29)	23.81 ^(3)^ (6.17)	25.67 ^(3)^ (6.63)	22.52 ^(3)^ (6.43)	23.56 ^(3)^ (7.04)	24.28 ^(3)^ (6.85)	23.25 ^(3)^ (5.84)	26.20 ^(3)^ (6.83)	27.15 ^(3)^ (6.80)	23.90 ^(3)^ (6.34)	25.54 ^(3)^ (6.71)	26.42 ^(3)^ (6.74)
	CS	37.63 ^(3)^ (7.81)	37.66 ^(3)^ (7.29)	38.33 ^(3)^ (7.32)	37.94 ^(3)^ (7.26)	37.00 ^(3)^ (7.72)	38.24 ^(3)^ (8.25)	29.67 ^(1)^ (8.17)	29.60 ^(1)^ (6.95)	33.81 ^(3)^ (6.99)	32.97 ^(3)^ (7.70)	31.79 ^(1)^ (7.92)	32.38 ^(2)^ (8.18)
OPR													
	PD	10.44 ^(5)^ (2.72)	11.31 ^(6)^ (2.80)	12.33 ^(7)^ (2.82)	11.65 ^(5)^ (2.65)	12.15 ^(7)^ (2.71)	12.59 ^(7)^ (2.95)	10.71 ^(5)^ (3.14)	12.02 ^(7)^ (2.98)	12.41 ^(7)^ (2.81)	10.02 ^(5)^ (3.35)	10.87 ^(5)^ (3.26)	11.60 ^(6)^ (3.45)
	CW-PD	7.02 ^(5)^ (3.35)	6.75 ^(5)^ (3.10)	6.98 ^(5)^ (3.35)	6.74 ^(5)^ (3.31)	6.94 ^(5)^ (3.49)	7.13 ^(5)^ (3.91)	9.89 ^(7)^ (4.32)	10.28 ^(7)^ (3.97)	9.68 ^(7)^ (4.25)	8.41 ^(5)^ (3.68)	8.65 ^(5)^ (3.85)	9.39 ^(6)^ (4.40)
	SS-QL	7.65 ^(7)^ (3.39)	7.52 ^(7)^ (3.25)	8.25 ^(7)^ (3.44)	7.36 ^(7)^ (3.48)	7.99 ^(7)^ (3.61)	8.48 ^(7)^ (3.96)	8.07 ^(7)^ (3.32)	7.83 ^(7)^ (3.88)	8.19 ^(7)^ (3.24)	7.83 ^(7)^ (3.75)	8.25 ^(7)^ (3.87)	8.29 ^(7)^ (3.87)
	C	4.10 ^(5)^ (2.52)	4.51 ^(5)^ (2.58)	5.12 ^(6)^ (2.92)	3.99 ^(5)^ (2.60)	4.55 ^(5)^ (2.52)	5.18 ^(6)^ (3.07)	4.68 ^(5)^ (2.80)	5.14 ^(6)^ (2.78)	5.89 ^(6)^ (2.96)	4.52 ^(5)^ (2.26)	5.23 ^(6)^ (2.92)	5.10 ^(6)^ (2.90)
	DP	2.36 ^(5)^ (1.67)	2.87 ^(5)^ (1.87)	3.58 ^(6)^ (2.11)	2.59 ^(5)^ (1.75)	2.84 ^(5)^ (1.76)	3.42 ^(6)^ (2.20)	2.54 ^(5)^ (1.82)	2.90 ^(5)^ (1.89)	3.34 ^(6)^ (1.93)	2.80 ^(5)^ (1.78)	3.37 ^(6)^ (1.95)	3.66 ^(6)^ (1.94)

SPS: Sensory Processing Sensitivity; PQoL: Professional Quality of Life: CF: Compassion Fatigue; BU: Burnout; CS: Compassion Satisfaction; OPR: Occupational Psychosocial Risks; PD: Psychological Demands; CW-PD: Control over Work and Possibilities of Development; SS-QL: Social Support and Quality Leadership; C: Compensations; DP: Double Presence; Levels in Sensory Processing Sensitivity—Spanish (HSPS-S) (see [31]): low ^(a)^: percentile < 34 [men = 27–140, women = 27–151]; medium ^(b)^: percentile 34 to 66 [men = 141–159, women = 152–167]; high ^(c)^: percentile >66 [men = 160–189, women = 168–189]. Level of exposure to indicators of professional quality of life (ProQoL): ^(1)^ low, ^(2)^ low–medium, ^(3)^ medium, ^(4)^ medium–high, ^(5)^ high. The high, medium and low score classification is: CF (low: 0–7; medium: 8–17; high: 18–50), BU (low: 0–17; medium: 18–27; high: 28–50) and CS (low: 0–32; medium: 33–41; high: 42–50). Level of exposure to occupational psychosocial risks (ISTAS21): ^(5)^ medium, ^(6)^ medium–high, ^(7)^ high. The high, medium and low score classification is: PD (low: 0–8; medium: 9–11; high: 12–20), CW-PD (low: 0–5; medium: 6–8; high: 9–20), SS-QL (low: 0–3; medium: 4–6; high: 7–20), C (low: 0–2; medium: 3–5; high: 6–12) and DP (low: 0–1; medium: 2–3; high: 4–8).

**Table 5 behavsci-13-00496-t005:** Comparison of means according to the level of SPS in each profession.

Education (*n* = 1190)									
		Comparison of Means		Paired Comparison of Means (t/U) SPS					
				Low–Medium		Low–High		Medium–High	
		F/H	*p*-value	*p*-value	Cohen’s d	*p*-value	Cohen’s d	*p*-value	Cohen’s d
PQoL									
	CF	159.090	0.000 ***	0.000 ***	−0.58(s)	0.000 ***	−1.25(h)	0.000 ***	−0.71(m)
	BU	30.977	0.000 ***	0.001 **	−0.28(s)	0.000 ***	−0.55(s)	0.000 ***	−0.29(s)
	CS	1.202	0.301	0.099	0.01(s)	0.392	−0.09(s)	0.385	−0.09(s)
OPR									
	PD	45.049	0.000 ***	0.000 ***	−0.31(s)	0.000 ***	−0.68(m)	0.000 ***	−0.36(s)
	CW-PD	1.283	0.534	0.526	0.08(s)	0.988	0.01(s)	0.559	−0.07(s)
	SS-QL	5.823	0.003 **	0.852	0.04(s)	0.036 *	−0.18(s)	0.004 **	−0.22(s)
	C	14.449	0.000 ***	0.079	−0.16(s)	0.000 ***	−0.37(s)	0.003 **	−0.22(s)
	DP	69.303	0.000 ***	0.000 ***	−0.29(s)	0.000 ***	−0.64(m)	0.000 ***	−035(s)
Healthcare (*n* = 914)									
		Comparison of means		Paired comparison of means (t/U) SPS					
				Low–Medium		Low–High		Medium–High	
		F/H	*p*-value	*p*-value	Cohen’s d	*p*-value	Cohen’s d	*p*-value	Cohen’s d
PQoL									
	CF	89.825	0.000 ***	0.000 ***	−0.49(s)	0.000 ***	−1.08(m)	0.000 ***	−0.59(s)
	BU	5.081	0.006 *	0.257	−0.15(s)	0.004 **	−0.26(s)	0.234	−0.10(s)
	CS	2.083	0.125	0.254	0.12(s)	0.888	−0.04(s)	0.151	−0.15(s)
OPR									
	PD	8.619	0.000 ***	0.063	−0.19(s)	0.000 ***	−0.33(s)	0.136	−0.15(s)
	CW-PD	0.666	0.719	0.752	−0.06(s)	0.396	−0.11(s)	0.807	−0.05(s)
	SS-QL	7.127	0.001 **	0.062	−0.18(s)	0.001 **	−0.30(s)	0.273	−0.13(s)
	C	14.345	0.000 ***	0.016 *	−0.22(s)	0.000 ***	−0.42(s)	0.020 *	−0.22(s)
	DP	23.152	0.000 ***	0.082	−0.14(s)	0.000 ***	−0.42(s)	0.002 **	−0.29(s)
Hospitality (*n* = 305)									
		Comparison of mean		Paired comparison of mean (t/U) SPS					
				Low–Medium		Low–High		Medium–High	
		F/H	*p*-value	*p*-value	Cohen’s d	*p*-value	Cohen’s d	*p*-value	Cohen’s d
PQoL									
	CF	37.502	0.000 ***	0.000 ***	−0.70(m)	0.000 ***	−1.25(h)	0.000 ***	−0.59(s)
	BU	8.641	0.000 ***	0.005 **	−0.46(s)	0.000 ***	−0.61(m)	0.532	−0.14(s)
	CS	11.000	0.000 ***	0.997	0.01(s)	0.000 ***	−0.54(s)	0.000 ***	−0.60(s)
OPR									
	PD	8.024	0.000 ***	0.006 **	−0.43(s)	0.000 ***	−0.57(s)	0.591	−0.13(s)
	CW-PD	0.600	0.741	0.789	−0.09(s)	0.936	0.05(s)	0.534	0.14(s)
	SS-QL	0.306	0.736	0.877	0.07(s)	0.972	−0.04(s)	0.726	−0.10(s)
	C	3.319	0.014 *	0.488	−0.17(s)	0.012 *	−0.42(s)	0.126	−0.26(s)
	DP	4.212	0.027 *	0.228	−0.19(s)	0.008 **	−0.43(s)	0.099	−0.23(s)
Administration/Management (*n* = 771)									
		Comparison of means		Paired comparison of means (t/U) SPS					
				Low–Medium		Low–High		Medium–High	
		F/H	*p*-value	*p*-value	Cohen’s d	*p*-value	Cohen’s d	*p*-value	Cohen’s d
PQoL									
	CF	72.306	0.000 ***	0.000 ***	−0.65(m)	0.000 ***	−1.07(m)	0.000 ***	−0.41(s)
	BU	9.668	0.000 ***	0.014 *	−0.25(s)	0.000 ***	−0.38(s)	0.014 *	−0.13(s)
	CS	1.498	0.224	0.194	0.15(s)	0.636	0.07(s)	0.667	−0.07(s)
OPR									
	PD	14.095	0.000 ***	0.015 *	−0.25(s)	0.000 ***	−0.26(s)	0.033 *	−0.46(s)
	CW-PD	8.745	0.033 *	0.593	−0.06(s)	0.014 *	−0.24(s)	0.068	−0.18(s)
	SS-QL	1.104	0.332	0.442	−0.11(s)	0.363	−0.12(s)	0.992	−0.01(s)
	C	4.616	0.010 *	0.011 *	−0.27(s)	0.040 *	−0.22(s)	0.871	−0.04(s)
	DP	24.270	0.000 ***	0.002 *	−0.30(s)	0.000 ***	−0.46(s)	0.081	−0.15(s)

*** *p* ≤ 0.001, ** *p* ≤ 0.01. * *p* ≤ 0.05. F = Fisher; t = Student; H = Kruskal–Wallis; U = Mann–Whitney. Cohen’s d: effect size: s = small magnitude ratio: <0.30; m = medium: 0.30–0.50; h = high: >0.50. SPS: Sensory Processing Sensitivity; PQoL: Professional Quality of Life: CF: Compassion Fatigue; BU: Burnout; CS: Compassion Satisfaction; OPR: Occupational Psychosocial Risks: PD: Psychological Demands; CW-PD: Control over Work and Possibilities of Development; SS-QL: Social Support and Quality Leadership; C: Compensations; DP: Double Presence.

**Table 6 behavsci-13-00496-t006:** Relationship between occupational psychosocial risks and professional quality of life in people with medium–high sensory processing sensitivity (*n* = 2192).

	Education (*n* = 862)			Healthcare (*n* = 582)			Hospitality (*n* = 221)			Administration/Management (*n* = 527)		
	CF	BU	CS	CF	BU	CS	CF	BU	CS	CF	BU	CS
PD	0.47 **(m) 0.000	0.50 ***(h) 0.000	−0.19 ***(s) 0.000	0.49 ***(m) 0.000	0.47 ***(m) 0.000	−0.18 ***(s) 0.000	0.49 **(m) 0.000	0.53 ***(h) 0.000	−0.24 ***(s) 0.000	0.49 ***(m) 0.000	0.55 ***(h) 0.000	−0.20 ***(s) 0.000
CW-PD	0.13 ***(s) 0.000	0.41 ***(m) 0.000	−0.53 ***(h) 0.000	0.20 ***(s) 0.000	0.44 ***(m) 0.000	−0.64 ***(h) 0.000	0.07 0.158	0.30 ***(m) 0.000	−0.56 ***(h) 0.000	0.07 *(s) 0.045	0.32 ***(m) 0.000	−0.55 ***(h) 0.000
SS-QL	0.31 ***(m) 0.000	0.48 ***(m) 0.000	−0.30 ***(s) 0.000	0.34 ***(m) 0.000	0.50 ***(h) 0.000	−0.35 ***(m) 0.000	0.25 ***(s) 0.000	0.36 ***(m) 0.000	−0.28 ***(s) 0.000	0.31 ***(m) 0.000	0.43 ***(m) 0.000	−0.34 ***(m) 0.000
C	0.34 ***(m) 0.000	0.39 ***(m) 0.000	−0.27 ***(s) 0.000	0.39 ***(m) 0.000	0.45 ***(m) 0.000	−0.28 ***(s) 0.000	0.39 ***(m) 0.000	0.42 ***(m) 0.000	−0.19 **(s) 0.003	0.35 ***(m) 0.000	0.41 ***(m) 0.000	−0.36 ***(m) 0.000
DP	0.38 **(m) 0.000	0.24 ***(s) 0.000	−0.03 0.163	0.30 ***(m) 0.000	0.21 ***(s) 0.000	−0.07 *(s) 0.041	0.35 ***(m) 0.000	0.32 ***(m) 0.000	−0.10 0.064	0.28 ***(s) 0.000	0.11 ***(s) 0.006	−0.01 0.446

*** *p* ≤ 0.001, ** *p* ≤ 0.01, * *p* ≤ 0.05. Effect size: small s = <0.30; medium m = 0.30–0.49; high h = >0.49. CF: Compassion Fatigue; BU: Burnout; CS: Compassion Satisfaction; PD: Psychological Demands; CW-PD: Control over Work and Possibilities of Development; SS-QL: Social Support and Quality Leadership; C: Compensations; DP: Double Presence.

**Table 7 behavsci-13-00496-t007:** Predictive factors related to occupational psychosocial risks on indicators of professional quality of life in the different sectors in people with medium–high sensory processing sensitivity (*n* = 2192).

Models														
Education (*n* = 862)														
Compassion fatigue							Burnout				Compassion satisfaction			
Models				R^2^	ΔR^2^	ChR^2^	Models	R^2^	ΔR^2^	ChR^2^	Models	R^2^	ΔR^2^	ChR^2^
1				0.223	0.222	0.223	1	0.252	0.251	0.252	1	0.279	0.278	0.279
2				0.304	0.303	0.082	2	0.358	0.356	0.106	2	0.297	0.296	0.019
3				0.339	0.336	0.034	3	0.407	0.405	0.049	3	0.302	0.299	0.004
4				0.343	0.34	0.004	4	0.428	0.425	0.021				
							5	0.441	0.438	0.013				
R^2^ = 34.3%							R^2^ = 44.1%				R^2^ = 30.2%			
F = 111.68; *p* = 0.000 *** df = 4.857							F = 134.82; *p* = 0.000 *** df = 5.856				F = 123.41; *p* = 0.000 *** df = 3.858			
Healthcare (*n* = 582)														
Compassion fatigue							Burnout				Compassion satisfaction			
Models				R^2^	ΔR^2^	ChR^2^	Models	R^2^	ΔR^2^	ChR^2^	Models	R^2^	ΔR^2^	ChR^2^
1				0.241	0.239	0.241	1	0.253	0.252	0.253	1	0.41	0.409	0.41
2				0.318	0.315	0.077	2	0.363	0.361	0.11	2	0.423	0.421	0.013
3				0.351	0.347	0.033	3	0.426	0.424	0.063				
							4	0.445	0.441	0.019				
							5	0.453	0.448	0.008				
R^2^ = 35.1%							R^2^ = 45.3%				R^2^ = 42.3%			
F = 104.02; *p* = 0.000 *** df = 3.578							F = 95.33; *p* = 0.000 *** df = 5.576				F = 212.31; *p* = 0.000 *** df = 2.579			
Hospitality (*n* = 221)														
Compassion fatigue							Burnout				Compassion satisfaction			
Models				R^2^	ΔR^2^	ChR^2^	Models	R^2^	ΔR^2^	ChR^2^	Models	R^2^	ΔR^2^	ChR^2^
1				0.241	0.237	0.241	1	0.285	0.281	0.285	1	0.316	0.312	0.316
2				0.294	0.288	0.053	2	0.343	0.337	0.058	2	0.342	0.336	0.026
3				0.331	0.322	0.037	3	0.376	0.368	0.034				
							4	0.398	0.398	0.022				
R^2^ = 33.1%							R^2^ = 39.8%				R^2^ = 34.2%			
F = 35.75; *p* = 0.000 *** df = 3.217							F = 35.69; *p* = 0.000 *** df = 4.216				F = 56.63; *p* = 0.000 *** df = 2.218			
Administration/Management (*n* = 527)														
Compassion fatigue							Burnout				Compassion satisfaction			
Models				R^2^	ΔR^2^	ChR^2^	Models	R^2^	ΔR^2^	ChR^2^	Models	R^2^	ΔR^2^	ChR^2^
1				0.239	0.238	0.239	1	0.307	0.306	0.307	1	0.299	0.298	0.299
2				0.285	0.282	0.045	2	0.371	0.369	0.064	2	0.327	0.325	0.028
3				0.317	0.313	0.032	3	0.394	0.391	0.023	3	0.335	0.331	0.007
							4	0.404	0.4	0.01				
R^2^ = 31.7%							R^2^ = 40.4%				R^2^ = 33.5%			
F = 80.88; *p* = 0.000 *** df = 3.523							F = 88.77; *p* = 0.000 *** df = 4.522				F = 87.91; *p* = 0.000 *** df = 3.523			
Predictive values														
Education (*n* = 862)														
Compassion fatigue					Burnout					Compassion satisfaction				
	β	Beta	t	*p*-value		β	Beta	t	*p*-value		β	Beta	t	*p*-value
Constant	6.191		6.545	0.000 ***	Constant	6.88		9.051	0.000 ***	Constant	48.051		77.985	0.000 ***
PD	0.946	0.352	11.649	0.000 ***	PD	0.746	0.329	11.801	0.000 ***	CW-PD	−1.061	−0.0470	−15.181	0.000 ***
DP	0.975	0.258	8.993	0.000 ***	CW-PD	0.494	0.247	8.867	0.000 ***	C	−0.298	−0.0113	−3.535	0.000 ***
C	0.457	0.166	5.266	0.000 ***	SS-QL	0.369	0.192	6.23	0.000 ***	SS-QL	−0.165	−0.076	−2.298	0.022 *
SS-QL	0.176	0.077	2.403	0.016 *	DP	0.409	0.128	4.828	0.000 ***					
					C	0.303	0.13	4.441	0.000 ***					
Healthcare (*n* = 582)														
Compassion fatigue					Burnout					Compassion satisfaction				
	β	Beta	t	*p*-value		β	Beta	t	*p*-value		β	Beta	t	*p*-value
Constant	2.983		2.399	0.017 *	Constant	4.57		4.45	0.000 ***	Constant	51.194		43.107	0.000 ***
PD	1.145	0.407	11.601	0.000 ***	SS-QL	0.362	0.197	5.004	0.000 ***	CW-PD	−1.359	−0.628	−19.780	0.000 ***
C	0.736	0.257	7.373	0.000 ***	PD	0.82	0.333	10.194	0.000 ***	PD	−0.328	−0.116	−3.665	0.000 ***
DP	0.746	0.185	5.399	0.000 ***	CW-PD	0.476	0.25	7.097	0.000 ***					
					C	0.379	0.153	4.05	0.000 ***					
					DP	0.31	0.089	2.821	0.005 **					
Hospitality (*n* = 221)														
Compassion fatigue					Burnout					Compassion satisfaction				
	β	Beta	t	*p*-value		β	Beta	t	*p*-value		β	Beta	t	*p*-value
Constant	8.987		4.731	0.000 ***	Constant	8.051		4.701	0.000 ***	Constant	46.049		24.538	0.000 ***
PD	0.93	0.243	5.545	0.000 ***	PD	0.904	0.385	6.536	0.000 ***	CW-PD	−0.954	−0.539	−9.712	0.000 ***
C	0.625	0.232	3.878	0.000 ***	C	0.472	0.199	3.412	0.001 **	PD	−0.410	−0.164	−2.955	0.003 **
DP	0.829	0.202	3.441	0.001 **	CW-PD	0.325	0.196	3.571	0.000 ***					
					DP	0.552	0.155	2.781	0.006 **					
Administration/Management (*n* = 527)														
Compassion fatigue					Burnout					Compassion satisfaction				
	β	Beta	t	*p*-value		β	Beta	t	*p*-value		β	Beta	t	*p*-value
Constant	8.334		7.765	0.000 ***	Constant	10.287		11.546	0.000 ***	Constant	45.134		40.147	0.000 ***
PD	0.915	0.39	10.032	0.000 ***	PD	0.886	0.074	11.986	0.000 ***	CW-PD	−0.935	−0.483	−12.612	0.000 ***
DP	0.824	0.203	5.542	0.000 ***	CW-PD	0.262	0.062	4.228	0.000 ***	C	−0.408	−0.147	−3.620	0.000 ***
C	0.521	0.191	4.946	0.000 ***	SS-QL	0.229	0.075	3.039	0.002 **	PD	−0.220	−0.092	−2.416	0.016 *
					C	0.286	0.096	2.974	0.003 **					

Models: *** *p* = 0.000; ΔR^2^ = R^2^ adjusted; ChR^2^ = Change in R^2^. Predictive values: *** *p* ≤ 0.001, ** *p* ≤ 0.01. * *p* ≤ 0.05. Occupational Psychosocial Risks: PD: Psychological Demands; CW-PD: Control over Work and Possibilities of Development; SS-QL: Social Support and Quality Leadership; C: Compensations; DP: Double Presence.

## Data Availability

The data presented in this article are available on request from the corresponding author.

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
