# Peer review of "Occupational Psychosocial Risks and Quality of Professional Life in Service Sector Workers with Sensory Processing Sensitivity"

_behavsci, 2023, doi:10.3390/bs13060496_

Round 1

Reviewer 1 Report

The study explores the role of sensory processing sensitivity in the perception of stress under different working conditions and its relationship with quality of professional life with a large sample survey in service sectors. The comparisons in the education, health care, hospitality, and administrative/management sectors have some practical value. However, the manuscript could be further improved in terms of research design and statistical validation.

1. In the introduction section, the authors are suggested to specify the reasons for the variables selected in the study. Why did you choose Psychological demands, Control over work and possibilities of development, Social support and quality leadership, Compensations, and Double presence salary as work condition? Why did you choose burnout, Compassion fatigue and Compassion satisfaction as quality of professional life.Every dimension should be clearly defined.

2. The relationship between certain dimensions of work condition such as control, compensation, etc. and dimensions of quality of professional life such as burnout mentioned in the study has been confirmed by numerous studies, and the authors should add  existing studies in the manuscript and clearly state the differences between this study and previous studies.

3. Perceived stress is mentioned repeatedly in the paper.In the measurement section authors mentioned the use of ISTAS 21 (CoPSoQ) Copenhagen Psychosocial Questionnaire to measure psychosocial risks in the job context, is this equivalent to perceived stress?

4. This study concerns the relationship between stressful working conditions and indicators of professional life in employees with high sensory processing sensitivity. It is suggested that the authors verify the existence of a moderating effect of sensory processing sensitivity in the relationship.

5. It is recommended that the authors develop a research hypothesis based on the relevant literature and draw a theoretical model.

6. There is some redundancy in the statistical analysis, especially for  the  two-by-two comparison in all the sectors. The authors could consider a more brief approach to verify  the relationship between work condition and quality of professional life. e.g. structural equations.

7. The conclusion section needs to specifically discuss the relationship between different work conditions and different qualities of professional life. The mechanism and direction of each dimension are not different, and generalizing the relationship has limited value.

8. Incomplete presentation of statistical tables.

Author Response

Response to Reviewer 1

Dear reviewer, thank you for making these suggestions that improve the manuscript substantially. We have taken note of them and have made the suggested changes. We hope that they will be to your liking.

The study explores the role of sensory processing sensitivity in the perception of stress under different working conditions and its relationship with quality of professional life with a large sample survey in service sectors. The comparisons in the education, health care, hospitality, and administrative/management sectors have some practical value. However, the manuscript could be further improved in terms of research design and statistical validation.

  1. In the introduction section, the authors are suggested to specify the reasons for the variables selected in the study. Why did you choose Psychological demands, Control over work and possibilities of development, Social support and quality leadership, Compensations, and Double presence salary as work condition? Why did you choose burnout, Compassion fatigue and Compassion satisfaction as quality of professional life.Every dimension should be clearly defined.

We have revised the introduction, in order to explain much more clearly the reasons for the selected variables. (lines 8 to 16, lines 28 to 33, lines 39 to 40, lines 82 to 90). We have highlighted the changes in yellow.

  1. The relationship between certain dimensions of work condition such as control, compensation, etc. and dimensions of quality of professional life such as burnout mentioned in the study has been confirmed by numerous studies, and the authors should add existing studies in the manuscript and clearly state the differences between this study and previous studies.

What differentiates this study from previous studies is to see how these relationships occur in workers according to their levels of sensitivity to sensory processing. The emphasis on the personality trait and how, depending on the presence of this trait, workers perceive more or less stress in different working conditions, in which sectors they have more difficulty and how it affects their quality of life is what interests you. Above all, in order to take these findings into consideration in the design of intervention strategies focused not only on the organisation but also on providing the necessary resources to workers who possess this trait, so that it is not a handicap for them and so that they can be able to make the most of their potential.

  1. Perceived stress is mentioned repeatedly in the paper.In the measurement section authors mentioned the use of ISTAS 21 (CoPSoQ) Copenhagen Psychosocial Questionnaire to measure psychosocial risks in the job context, is this equivalent to perceived stress?

ISTAS 21 (CoPSoQ) assesses perceived stress under different working conditions, which are those indicated in the study. That is, a worker may perceive stress due to perceived work overload, or perceived lack of support, etc.... Therefore, in the case of this study, when we talk about perceived stress we are referring to the stress that workers are experiencing.

  1. This study concerns the relationship between stressful working conditions and indicators of professional life in employees with high sensory processing sensitivity. It is suggested that the authors verify the existence of a moderating effect of sensory processing sensitivity in the relationship.

The purpose of this study was to determine the existing differences in the perception of stress in certain working conditions according to the levels of sensitivity to sensory processing, i.e., whether there were differences depending on whether they had high, medium or low levels. Furthermore, once it has been verified that there are indeed differences, from the initial sample, linear regression analyses are carried out only with the workers who present high levels, since what we are interested in is finding out how perceived stress (by these workers in particular) affects quality of life indicators. The emphasis is again on the highly sensitive workers. The aim of this study is not to find out the moderating role of the trait but how stress affects the quality of life of people who have the trait and who work in different sectors of activity according to which the perception of stress under different working conditions may vary in the general population, but in this study we are interested in what happens to people who have the trait of sensory processing sensitivity.

  1. It is recommended that the authors develop a research hypothesis based on the relevant literature and draw a theoretical model.

We have revised the introduction, in order to better explain the working hypothesis and highlight the relevant literature. (line 8 to 16, line 28 to 33, line 39 to 40, line 82 to 90). We have highlighted the changes in yellow.

  1. There is some redundancy in the statistical analysis, especially for the two-by-two comparison in all the sectors. The authors could consider a more brief approach to verify  the relationship between work condition and quality of professional life. e.g. structural equations.

As mentioned, firstly, the aim is to see the differences between sectors in the different variables studied: stress in different working conditions, sensory processing sensitivity (SPS), compassion fatigue, burnout and compassion satisfaction. Secondly, the aim is to find out, within each sector, what variations there are in the experience of stress in working conditions and in the indicators of quality of working life (compassion fatigue, burnout and compassion satisfaction) according to the level of workers' sensitivity to sensory processing (low, medium or high SPS). We consider the reviewer's suggestion to be very interesting and, in fact, it is part of another study in which the authors are working on the mediating role of Sensory Processing Sensitivity, however, we understand that in this work the analyses carried out are appropriate to respond to the initial objectives, focusing more on the differences than on the moderating or mediating role.

  1. The conclusion section needs to specifically discuss the relationship between different work conditions and different qualities of professional life. The mechanism and direction of each dimension are not different, and generalizing the relationship has limited value.

We have improved the introduction to explain in more detail the relationship between variables (lines 8 to 16, lines 28 to 33, lines 39 to 40, lines 82 to 90). We have highlighted the changes in yellow.

  1. Incomplete presentation of statistical tables.

We have corrected the orientation of the page that prevented the correct display of the data.

Reviewer 2 Report

Dear authors

Thank you so much for your work. The manuscript presents an interesting topic. However, in the current state, the study needs to be revised. I present my main concerns:

1-     The introduction section should be thought of as the “visiting card” of the study, making the purpose and contribution of the investigation clear. However, in its current state, the introduction does not serve its purpose given that it is not entirely clear what this work adds to the literature, why it is essential, what is already known in general terms, and how authors intend to contribute. On the other hand, The authors mentioned the following:

“On the other hand, the literature does not show any studies on other service sector professions in which suffering is associated with other needs, such as well-being (nutrition and rest) (Hospitality sector) and the management of economic matters (Administration/management sector), which is an element of interest for the authors of the present study”. I argue that justifying the study based on its scarcity is not enough. Additionally, mentioning that an investigation was carried out because it was in the interest of the researchers is an argument that should be reformulated because research should be carried out to improve the theoretical and practical understanding of a specific topic, and not to address the author's interest.

2-Authors should present a literature review section with the definition of the main variables under study (Including the definition of subdimensions), as well as a more robust theoretical foundation of the established relationships.

3-Although the authors used the Spanish versions of the scales, the process of validating the scales is a crucial issue. Therefore, I suggest that the authors present the results of the confirmatory factor analysis of all the scales used.

4- The authors decided to maintain subscales with Cronbach's alpha values below 0.70 without presenting a theoretical basis. Therefore, I argue that this issue should be addressed. On the other hand, the authors did not present the correlations between the main variables.

5- The authors did not mention the procedures used to minimize the common method bias. This is a critical issue that should be addressed.

6- The presented limitations should be reformulated for the following reasons:

a)     The use of the short version of the questionnaires is an author's option. Therefore, the limitation should be, for example, the difficulty in generalizing the results of this study.

b)     The online application of the questionnaires is also an author's option. According to this reasoning, this limitation should be covered for all those who did not complete the questionnaire for other reasons.

However, the real limitations were not addressed. For example, the fact that it is a cross-sectional study using a single source of data collection; the fact that this study presents only a quantitative perspective in the analysis of the results.

Author Response

Response to Reviewer 2

Dear reviewer, thank you for making these suggestions that improve the manuscript substantially. We have taken note of them and have made the suggested changes. We hope that they will be to your liking.

Thank you so much for your work. The manuscript presents an interesting topic. However, in the current state, the study needs to be revised. I present my main concerns:

1-     The introduction section should be thought of as the “visiting card” of the study, making the purpose and contribution of the investigation clear. However, in its current state, the introduction does not serve its purpose given that it is not entirely clear what this work adds to the literature, why it is essential, what is already known in general terms, and how authors intend to contribute. On the other hand, The authors mentioned the following:

“On the other hand, the literature does not show any studies on other service sector professions in which suffering is associated with other needs, such as well-being (nutrition and rest) (Hospitality sector) and the management of economic matters (Administration/management sector), which is an element of interest for the authors of the present study”. I argue that justifying the study based on its scarcity is not enough. Additionally, mentioning that an investigation was carried out because it was in the interest of the researchers is an argument that should be reformulated because research should be carried out to improve the theoretical and practical understanding of a specific topic, and not to address the author's interest.

The main objective of this work is the study of people with sensory processing sensitivity (therefore, of relevance to the study of clinical psychology), and how these people with this personality trait experience the experience related to work performance or work on a daily basis. Certainly, as the reviewer of the manuscript points out, the relationship between working conditions, stress and its consequences has been addressed on numerous occasions in the existing literature, with contributions from work and organisational psychology standing out especially and effectively. However, in addition to the organisation, the study of people is of special relevance, and in this specific case, the aim is to contribute with a pioneering contribution on people with high sensitivity. Hence the relevance of this study.

In relation to the reviewer's contribution that research carried out in the interest of researchers is an argument to be reformulated, we absolutely agree that it aspires to a more ambitious scope. All the effort, dedication and resources made by the researchers, in general, and in particular the authors of this work, is to contribute to knowledge, understood from a theoretical and applied perspective, which in this case would translate into a reference for professionals in clinical psychology and clinical psychology at work, through individual interventions, as well as for professionals in work psychology, through proposals for preventive actions for people with high sensitivity. Mental health complications are particularly relevant to address, for which it is essential to know and understand how this population group experiences working conditions. With this perspective, both the theoretical framework and the methodological design have been designed, although it is understandable and very respectable that professionals with a different approach to that of the authors of this work consider another approach to the approach of this work.

On the other hand, in response to the text in inverted commas, understanding these people with a high level of empathy is fundamental, because unlike people without this personality trait, the stress they experience in the face of other people's suffering and/or discomfort is notably higher and disabling. Hence the authors' interest in finding out more about how people, especially those with high sensitivity, are similar and different, in order to contribute these findings to the knowledge of the scientific community.

2-Authors should present a literature review section with the definition of the main variables under study (Including the definition of subdimensions), as well as a more robust theoretical foundation of the established relationships.

We have further substantiated the established relationships, adding relevant literature and clarifying the relationship between variables. (lines 8 to 16, lines 28 to 33, lines 39 to 40, lines 82 to 90). We have highlighted the changes in yellow.

3-Although the authors used the Spanish versions of the scales, the process of validating the scales is a crucial issue. Therefore, I suggest that the authors present the results of the confirmatory factor analysis of all the scales used.

Below is the reference information where the versions of the adaptations of the instruments used in this study can be found:

  • (pages 32 and 33) Cano, F. J., Rodríguez, L., y García, J. (2007). Spanish version of the Coping Strategies Inventory. Actas Españolas De Psiquiatría, 35(1), 29-39. https://www.researchgate.net/publication/235419673

  • (pages 1045-1046) Chacón, A., Pérez-Chacón, M., Borda-Mas, M., Avargues-Navarro, M.L. y López-Jiménez, A.M (2021). Cross-cultural adaptation and validation of the Highly Sensitive Person Scale to the adult Spanish population (HSPS-S). Psychology Research and Behavioral Management, 14, 1041-1052. DOI: 10.2147/PRBM.S321277

  • Moncada, S.; Llorens, C.; Navarro, A.; Kristensen, T. S. ISTAS21: The Spanish version of the Copenhagen psychosocial questionnaire (COPSOQ). ISTAS21: Versión en lengua castellana del cuestionario psicosocial de Copenhague (COPSOQ)]. Archivo de Prevención de Riesgos Laborales, 2005, 8(1), 18–29. https://onlinelibrary.wiley.com/doi/10.1002/ajim.22238

4- The authors decided to maintain subscales with Cronbach's alpha values below 0.70 without presenting a theoretical basis. Therefore, I argue that this issue should be addressed. On the other hand, the authors did not present the correlations between the main variables.

The authors are aware of the guidelines required for internal consistency values for both Cronbach's Alpha and McDonald's omega coefficient (acceptable from .70 and questionable from .60). Despite these guidelines, the existing literature also states that even if questionable is low (e.g. Lowenthal, 1996), values above .60 can be considered acceptable when the number of items to be analysed is less than 10. In addition, Kline (2015) takes into account the correlations between items, considering these values acceptable when there are no low correlations between items.

In this sense, bivariate correlations have been carried out, with the items that make up each of the scales / dimensions / factors, obtaining the following results:

FPD (HSPS-S): (4 items) all correlations are significant (p<0.001).

BU (ProQoL-vIV) (10 items), all correlations are significant (p<0.001) except it 15 with it 1 (p=0.013), it 4 (p=0.003), it 19 (p=0.019) and it 26 (p=0.027). There is no relationship between it 4 with it 19 and it 21 and it 17 with it 29.

PD (CoPSoQ) (5 items) all correlations are significant (p<0.001), except it1 with it 2 and i t5).

CW-PD (CoPSoQ) (5 items) all correlations significant (p<0.001), except it7 with it 9 and it 10) (p<0.01)

C (CoPSoQ) (3 items) all correlations are significant (p<0.001)

DP (CoPSoQ) (2 items) correlation r pearson=0.45 (p<0.001)

We have also incorporated the values of the McDonald's omega coefficients, in which some differences are observed (increasing internal consistency), values that have been inserted in the manuscript (page 5)

  • High-Sensitivity Person Scale (HSPS): “In the present study, the Cronbach’s α and McDonald´s Omega coefficients for the differ-ent subscales were: α = 0.87 and ω = 0.87 for sensitivity overstimulation”
  • ISTAS 21 (CoPSoQ): “In this study, the Cronbach’s α and McDonald´s Omega coefficients for the subscales were: PD (α = 0.58; ω = 0.71), CW-PD (α = 0.64; ω = 0.69), SS-QL (α = 0.70; ω = 0.75), C (α = 0.55; ω = 0.62) and DP (α = 0.64; ω = 0.65)”.
  • Spanish Adaptation of the Professional Quality of Life Scale (ProQoL-vIV): The reliability of the scale was 0.71 and the internal consistency coefficients were: CF (α = 0.80; ω = 0.81), CS (α = 0.85; ω = 0.85) and BU (α = 0.64; ω = 0.67).

5- The authors did not mention the procedures used to minimize the common method bias. This is a critical issue that should be addressed.

A concern that arises for authors conducting research studies with single-source self-report cross-sectional designs is common method bias, where observed relationships may be biased (Schaller et all, 2015).

Therefore, we have followed the indications found in the scientific literature to avoid common method bias, such as Podsakoff et all, 2012.

References

  • Kothandapani, V. (1971). Validation of feeling, belief, and intention to act as three components of attitude and their contribution to prediction of contraceptive behaviors. Journal of Personal Social Psychology, 19, 321-333.
  • Podsakoff, P.M., MacKenzie, S.B., & Podsakoff, N.P. (2012). Sources of method bias in social science research and recommendations on how to control it. Annual Review Psychology, 63, 539-569.
  • Schaller, T.K., Patil, A., & Malhotra, N.K. (2015). Alternative techniques for assessing common method variance: An analysis of the Theory of Planned Behavior research. Organizational Research Methods, 18, 177-206.
  • Siemsen, E., Roth, A., & Oliveira, P. (2010). Common method bias in regression models with linear, quadratic, and intraction effects. Organizational Research Methods, 13, 456-476.
  • Williams, L.J., Hartman, N., & Cavazotte, F. (2010). Method variance and marker variables: A review and comprehensive CFA marker technique. Organizational Research Methods, 13, 477-514.

6- The presented limitations should be reformulated for the following reasons:

  1. a) The use of the short version of the questionnaires is an author's option. Therefore, the limitation should be, for example, the difficulty in generalizing the results of this study.

  1. b) The online application of the questionnaires is also an author's option. According to this reasoning, this limitation should be covered for all those who did not complete the questionnaire for other reasons.

However, the real limitations were not addressed. For example, the fact that it is a cross-sectional study using a single source of data collection; the fact that this study presents only a quantitative perspective in the analysis of the results.

We have addressed the limitations of the study (section: Conclusions, Lines: 16 to 36). We have highlighted the changes in yellow.

Reviewer 3 Report

This manuscript presents a comprehensive and rigorous study on the topic of occupational psychosocial risks and quality of professional life in service sector workers. The authors have conducted a large-scale survey with a representative sample of workers from different service sectors, and have applied appropriate statistical methods to analyze the data.

The results are clearly reported and discussed, and the implications for practice and policy are well articulated. The manuscript is well written, organized and formatted, and follows the journal's guidelines. The literature review is thorough and up-to-date, and the references are relevant and accurate.

The manuscript makes a significant contribution to the field of occupational health psychology, and I recommend it for publication without any major revisions.

Author Response

Response to Reviewer 3

This manuscript presents a comprehensive and rigorous study on the topic of occupational psychosocial risks and quality of professional life in service sector workers. The authors have conducted a large-scale survey with a representative sample of workers from different service sectors,  and have applied appropriate statistical methods to analyze the data.

The results are clearly reported and discussed, and the implications for practice and policy are well articulated. The manuscript is well written, organized and formatted, and follows the journal's guidelines. The literature review is thorough and up-to-date, and the references are relevant and accurate.

The manuscript makes a significant contribution to the field of occupational health psychology, and I recommend it for publication without any major revisions.

Dear reviewer, thank you for making these suggestions. We have made some changes suggested by other reviewers that we hope you will like.

Round 2

Reviewer 1 Report

In general, the manuscript has been improved. But I still suggest that the paper should present clear research hypotheses, while the existing tables should be simplified.

Author Response

Response to Reviewer 1 ROUND 2

Dear reviewer, thank you for offering us improvement guidelines. Thanks to his help the manuscript has improved substantially. We now explain our working hypothesis in more detail.

In general, the manuscript has been improved. But I still suggest that the paper should present clear research hypotheses, while the existing tables should be simplified.

Again in response to their suggestions, and trying not to make substantial changes that would go against the suggestions of the other reviewers, we have incorporated the following text after the objectives:

Based on the objectives of this study, it is expected that exposure to certain working conditions may pose a risk to the quality of working life of workers in various service sector occupations. Specifically, it is expected that high psychological demands, lack of social support and quality leadership, and difficulty in finding a work-family balance act as risk factors, while control at work and development possibilities at work play a protective role. In addition, the presence of high sensitivity is expected to be associated with a higher presence of the negative dimensions of quality of work life (burnout and compassion fatigue). (Lines 82 - 89, introduction section)

Similarly, at the beginning of the discussion, the following sentence has been incorporated:

Personality traits and job characteristics are important factors in determining individual health status in the workplace [35]. The general objective of this study was to analyze the role of the SPS in the perception of stress in certain working conditions and its relationship with indicators of quality of professional life. The novelty of this study lies in the fact that it was carried out simultaneously with workers from different professions in the service sector, specifically Education, Health, Hospitality and Administration/management. The results obtained have allowed us to confirm our initial assumptions. (lines 7 – 8 discussion section).

With regard to the suggestion to simplify the tables, the authors consider that eliminating the information that appears in the tables or part of the information could lead to the loss of information necessary to meet the objectives of the study, as well as an important methodological limitation, by eliminating the information that appears in the tables.

Thus, for example, Table 3, for comparison of means, includes both the information necessary to determine whether there are differences between the four groups considered globally (including the test value and p-value) and then the pairwise comparisons between the four groups (including p-value, Cohen's d and effect size). Finally, it is detailed which of the two groups compared indicates a higher or lower score. The researchers believe that providing this data is critical to the understanding of the study.

From Table 4, the aim is not only to detail the relationships between the variables, but also to do so in terms of the level of SPS in each of the professions, taking into account the level of professional quality of life and the potential risk of the different working conditions. And finally, in Table 7, determining the predictive value of the different variables in the different professions allows us to identify not only the relationships but also their specific weight on the indicators of quality of professional life. From a care or preventive point of view, this highly detailed information allows for a better understanding of the phenomenon under study.

Eliminating tables or part of them could go against the suggestions made by the other reviewers, as information that they have considered to be of interest could be lost. If you consider it very relevant to make this modification, we would be grateful if you could indicate the exact information that you consider should be removed from the tables. We understand that, in this case, we could leave the final decision on how to proceed in this regard to the Editor, so as not to conflict with the suggestions of the other reviewers.

Reviewer 2 Report

Dear authors,

I really appreciate the effort taken to improve the manuscript. However, I could not clearly understand the procedures followed by the authors to minimize the common method bias.

Author Response

Dear Reviewer:

Please find attached a file with the detailed response to your observations. Thank you
